# GP-delivered medication review of polypharmacy, deprescribing, and patient priorities in older people with multimorbidity in Irish primary care (SPPiRE Study): A cluster randomised controlled trial

**Caroline McCarthy**[1]*, **Barbara Clyne**[1], **Fiona Boland**[1,2], **Frank Moriarty**[1,3], **Michelle Flood**[3], **Emma Wallace**[1], **Susan M. Smith**[1], **for the SPPiRE Study team**[¶]

**1** HRB Centre for Primary Care Research, Department of General Practice, Royal College of Surgeons in Ireland, University of Medicine and Health Sciences, Dublin, Ireland, **2** Data Science Centre, Royal College of Surgeons in Ireland, University of Medicine and Health Sciences, Dublin, Ireland, **3** School of Pharmacy and Biomolecular Sciences, Royal College of Surgeons in Ireland, University of Medicine and Health Sciences, Dublin, Ireland

¶ Membership of the SPPiRE Study team is listed in the Acknowledgements.

* carolinemccarthy@rcsi.ie

## Abstract

### Background

There is a rising prevalence of multimorbidity, particularly in older patients, and a need for evidence-based medicines management interventions for this population. The Supporting Prescribing in Older Adults with Multimorbidity in Irish Primary Care (SPPiRE) trial aimed to investigate the effect of a general practitioner (GP)-delivered, individualised medication review in reducing polypharmacy and potentially inappropriate prescriptions (PIPs) in community-dwelling older patients with multimorbidity in primary care.

### Methods and findings

We conducted a cluster randomised controlled trial (RCT) set in 51 GP practices throughout the Republic of Ireland. A total of 404 patients, aged ≥65 years with complex multimorbidity, defined as being prescribed ≥15 regular medicines, were recruited from April 2017 and followed up until October 2020. Furthermore, 26 intervention GP practices received access to the SPPiRE website where they completed an educational module and used a template for an individualised patient medication review that identified PIP, opportunities for deprescribing, and patient priorities for care. A total of 25 control GP practices delivered usual care. An independent blinded pharmacist assessed primary outcome measures that were the number of medicines and the proportion of patients with any PIP (from a predefined list of 34 indicators based predominantly on the STOPP/START version 2 criteria). We performed an intention-to-treat analysis using multilevel modelling. Recruited participants had substantial disease and treatment burden at baseline with a mean of 17.37 (standard deviation [SD]

**Data Availability Statement:** The SPPiRE dataset is available from the Zenodo repository (DOI: https://doi.org/10.5281/zenodo.5539817).

**Funding:** This research is funded by the HRB Primary Care Clinical Trial's Network, Ireland (https://primarycaretrials.ie/) HRB PC CTNI grant code 2029 awarded to SMS. BC is funded by a Health Research Board (HRB) Emerging Investigator Award (EIA-2019-09). EW is funded by a Health Research Board Ireland Emerging Clinician Scientist Award-ECSA-2020-02. The funders had no role in study design, data collection and analysis, decision to publish, or preparation of the manuscript.

**Competing interests:** The authors have declared that no competing interests exist.

**Abbreviations:** ADWE, adverse drug withdrawal event; CI, confidence interval; CONSORT, Consolidated Standards of Reporting Trials; COVID-19, Coronavirus Disease 2019; ED, emergency department; EQ-5D-5L, EuroQoL 5-dimension 5-level; GP, general practitioner; ICC, intraclass correlation; IQR, interquartile range; IRR, incidence rate ratio; MTBQ, Multimorbidity Treatment Burden Questionnaire; OPTI-SCRIPT, Optimizing Prescribing for Older People in Primary Care, a cluster-randomized controlled trial; OR, odds ratio; PINCER, Pharmacist-Led Information Technology Intervention for Medication Errors; PIP, potentially inappropriate prescription; PMS, practice management system; PPI, patient and public involvement; PRIMUM, Prioritising Multimedication in Multimorbidity; RCT, randomised controlled trial; rPATD, revised Patients' Attitudes Towards Deprescribing; SD, standard deviation; SNRI, serotonin–norepinephrine reuptake inhibitor; SPPiRE, Supporting Prescribing in Older Adults with Multimorbidity in Irish Primary Care; SSRI, selective serotonin reuptake inhibitor.

3.50) medicines. At 6-month follow-up, both intervention and control groups had reductions in the numbers of medicines with a small but significantly greater reduction in the intervention group (incidence rate ratio [IRR] 0.95, 95% confidence interval [CI]: 0.899 to 0.999, $p = 0.045$). There was no significant effect on the odds of having at least 1 PIP in the intervention versus control group (odds ratio [OR] 0.39, 95% CI: 0.140 to 1.064, $p = 0.066$). Adverse events recorded included mortality, emergency department (ED) presentations, and adverse drug withdrawal events (ADWEs), and there was no evidence of harm. Less than 2% of drug withdrawals in the intervention group led to a reported ADWE. Due to the inability to electronically extract data, primary outcomes were measured at just 2 time points, and this is the main limitation of this work.

## Conclusions

The SPPiRE intervention resulted in a small but significant reduction in the number of medicines but no evidence of a clear effect on PIP. This reduction in significant polypharmacy may have more of an impact at a population rather than individual patient level.

## Trial registration

ISRCTN Registry ISRCTN12752680.

## Author summary

### Why was this study done?

- Polypharmacy is frequently cited as a major concern by patients with multimorbidity.

- More evidence-based medicines management interventions are needed to improve care for this growing and vulnerable population.

### What did the researchers do and find?

- We conducted a pragmatic, 2-arm cluster randomised controlled trial (RCT) in Irish primary care to investigate whether a general practitioner (GP)-delivered individualised medication review with a deprescribing approach could reduce polypharmacy and improve prescribing in older people with multimorbidity and significant polypharmacy.

- There was a small but significant reduction in the number of medicines in the intervention compared to control group at follow-up but no significant effect on potentially inappropriate prescribing (PIP).

- Out of 826 medicines stopped in the intervention group, just 15 adverse drug withdrawal events (ADWEs) were reported.

**What do these findings mean?**

- A primary care–based medication review intervention that aims to reduce significant polypharmacy is safe and could lead to the deprescription of unnecessary medicines. However, this may have a greater impact at a population rather than individual patient level.

- Improvements in the control group suggest that identification of patients with significant polypharmacy may in itself lead to reduction in the number of medicines.

- Recruitment and retention of patients with a high degree of disease and treatment burden into RCTs are possible but require significant resource and planning, and targeting those with less severe disease burden may be more appropriate and may lead to greater gains for individual patients.

## Introduction

Advances in healthcare and therapeutics have led to increases in life expectancy and a rising population of older people living with multiple long-term conditions or multimorbidity. Multimorbidity, commonly defined as 2 or more chronic conditions, is associated with adverse outcomes for patients including increased mortality and reduced quality of life [1–3]. As numbers of conditions rise, there is increased burden on healthcare systems related to increased healthcare utilisation, particularly unplanned hospital admissions [4], which in older people are frequently the result of adverse drug reactions [5]. Polypharmacy is the single biggest risk factor for potentially inappropriate prescribing (PIP), which describes suboptimal prescribing where the risks of treatment generally outweigh the benefits [6–8]. In patients with multimorbidity, a degree of polypharmacy can be expected and an excess of unplanned hospital admissions is only seen in people with higher levels of polypharmacy [9]. Focusing on higher levels of polypharmacy ($\geq$15 medicines) is one approach to identifying those at risk of adverse drug-related events. Another approach, given that this group is also more likely to have PIP, is to assess the quality of prescribing. Explicit or implicit measures of medication appropriateness that have been developed through literature reviews and consensus methods are often employed for this purpose [10].

Prescribing for patients with multimorbidity is particularly complex, and, although explicit measures are useful in identifying potential drug–drug and drug–disease interactions, consideration must be given to the individual context and patient priorities [6]. This individualised approach is supported by multimorbidity and polypharmacy guidelines [11–15] and is preferred to a single disease focus, which can lead to an unacceptable treatment burden, defined as the work required of a patient to manage their medical conditions and includes taking medicines [16]. In addition, the actual effectiveness of many therapeutic interventions in older patients with multimorbidity is unclear, with these people typically excluded from the randomised controlled trial (RCTs) that are the basis for these single disease guidelines [6]. In recent years, there has been an increasing focus on the concept of deprescribing, described as "the process of withdrawal of an inappropriate medication, supervised by a health care professional with the goal of managing polypharmacy and improving outcomes" [17]. For many older patients, the risk benefit ratio of a particular therapeutic intervention may no longer be favourable. Deprescribing is an important and necessary component of effective prescribing and

involves an ongoing assessment of both the effectiveness and risks of treatment and patient preferences [18].

The development of the Supporting Prescribing in Older Adults with Multimorbidity in Irish Primary Care (SPPiRE) intervention has been extensively described elsewhere [19]. In summary, SPPiRE evolved from our research group's previous intervention (Optimizing Prescribing for Older People in Primary Care, a cluster-randomized controlled trial, OPTI-SCRIPT) [20] and incorporated emerging evidence in the fields of multimorbidity and polypharmacy, particularly the concepts of treatment burden and deprescribing. Medication count is a simple objective measure that has been identified as a good proxy measure for multimorbidity, predicting both primary healthcare usage and mortality [21], and, although treatment for a single condition could reach or exceed 5 medications, ≥15 repeat medicines is likely highly specific to multimorbidity. Based on recommendations from the United Kingdom NICE multimorbidity guidelines, which recommends targeting patients on 15 or more medicines for a multimorbidity approach to care and in keeping with the fact that higher levels of polypharmacy are more likely to be associated with adverse outcomes for patients with multimorbidity, the target population was patients aged ≥65 years, prescribed ≥15 repeat medicines [14]. The aim of the SPPiRE cluster RCT was to assess the effectiveness of a complex intervention that was targeted at general practitioners (GPs) and incorporated professional training and an individualised web-guided medication review that addressed the quality of prescribing and incorporated an assessment of patient treatment priorities, in reducing polypharmacy and PIP in older adults with significant polypharmacy and multimorbidity in Irish primary care.

## Methods

### Study design and participants

The methods for the SPPiRE cluster RCT have been described in the trial protocol [22] (S1 Text). This study is reported in line with the Consolidated Standards of Reporting Trials (CONSORT) 2010 cluster RCT checklist [23] (see S1 Table) and was approved by the Irish College of General Practitioners Research Ethics Committee. In brief, SPPiRE was a pragmatic, 2-arm cluster RCT, with the intervention delivered to GP clusters and analysis of outcomes at the patient level. Information about the trial was publicised through a variety of GP research, teaching, and training networks throughout Ireland. Eligible practices expressing an interest were formally invited. Practices were eligible to participate if they had at least 300 registered patients aged ≥65 years (based on the need to identify a sufficient number of eligible participants) and used either of the 2 Irish GP practice management systems (PMSs) with over 80% national cover; this enabled the use of a SPPiRE patient finder tool that was developed and embedded into these systems. Practices were excluded if they were currently involved in a medication management or prescribing trial or if they were unable to recruit at least 5 participants.

Eligible patients were aged ≥65 years and prescribed ≥15 repeat medicines. A repeat medicine was defined as any unique item with a World Health Organization Anatomical Therapeutic Chemical code on the patient's current repeat prescription, i.e., indicated by their GP as being an ongoing treatment, for ease of issuing further prescriptions within the patient record. Patients were excluded if they had been recruited into a practice that was unable to recruit at least 4 other participants, they were judged by their GP as unable to give informed consent, or they were unable to attend the practice for a face-to-face medication review (e.g., nursing home residents and housebound patients). Recruited GPs ran the SPPiRE patient finder tool and screened the generated list to ensure that only eligible patients were invited. Practices who

identified more than 40 eligible patients were supported in selecting a random sample of 30 patients to invite. All recruited practices and patients gave fully informed consent, and baseline data were collected prior to practice allocation, to reduce the likelihood of selection bias.

### Randomisation and masking

Recruited practices were allocated to intervention (26) or control groups (25) by minimisation using MinimPy software [24] by the trial statistician (FB) who had no knowledge of participating practices. Minimisation variables included practice size (number of GP sessions per week: 0 to 14, 14 to 28, and 28 or more) and location (urban, rural, or mixed). Considering the nature of the intervention, it was not possible to blind GPs or patients to the intervention; however, to reduce the risk of detection bias, the 2 primary outcome measures, the number of repeat medicines, and whether a PIP was present were assessed by an independent blinded pharmacist (MF).

### Procedures

Intervention GPs received unique log-in details to the SPPiRE website where they had access to 5 training videos and a template for performing the SPPiRE medication review. The training videos provided background information on multimorbidity and polypharmacy, PIP, eliciting patient treatment priorities, and conducting a brown bag medication review. GPs were instructed to book a double appointment and to ask their patients to bring all their medicines in to the medication review visit with them. The SPPiRE medication review process had 2 main components: gather and record information and then to discuss and agree upon any changes with their patient based on the recorded information, with a focus on deprescribing medicines that were potentially inappropriate (Fig 1). The website provided suggested treatment alternatives for identified PIP, but all treatment decisions were ultimately at the discretion of the individual GP, based on their clinical judgement and their patients' individual priorities.

Control GPs delivered usual care during the 6- to 12-month study period. At the time of intervention delivery, there was no structured chronic disease management programme in Irish primary care, and many patients with multimorbidity attended multiple hospital specialists. In Ireland, the majority of people aged 70 years of age have access to free GP visits and medicines with some prescription charge co-payments. In the 65- to 69-year-old age category, a lower proportion have access to both free GP visits and prescription medicines. Access to specialists and diagnostics in secondary care is free for the entire population.

### Outcomes

The 2 primary outcomes were the number of repeat medicines and the proportion of patients with any PIP, from a list of 34 prespecified indicators (see S2 Table). A series of secondary prescribing-related outcome measures was prespecified to allow a more in-depth analysis of the effect of the intervention on prescribing. These were the following:

- the number of medicines stopped and started;

- the proportion of patients with a reduction in significant polypharmacy (defined as ≥15 repeat medicines);

- the number of PIP;

- the proportion of patients with a high-risk PIP (see S2 Table); and

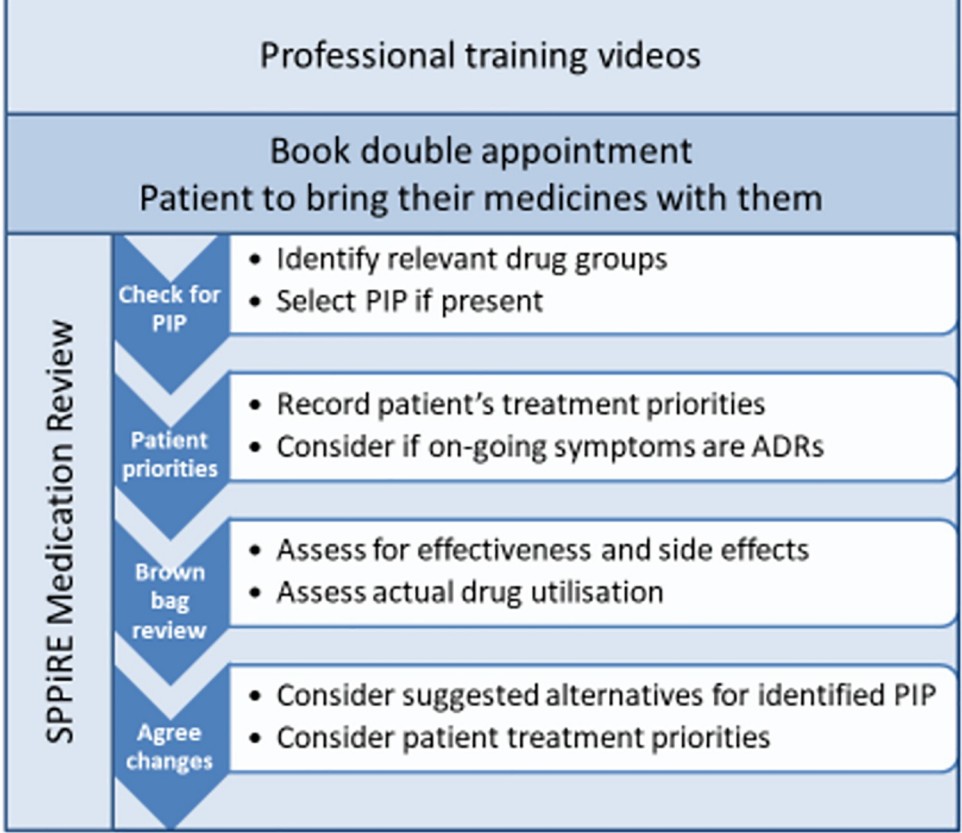

**Fig 1. SPPiRE intervention.** ADR, adverse drug reaction; GP, general practitioner; PIP, potentially inappropriate prescribing; SPPiRE, Supporting Prescribing in Older Adults with Multimorbidity in Irish Primary Care.

- the proportion of patients with any reduction in PIP.

Secondary patient-reported outcomes measures were included to capture the effectiveness of the intervention from the patients' perspective. These were the following:

- Health-related Quality of life (EuroQoL 5-dimension 5-level, EQ-5D-5L) [25];

- revised Patients' Attitudes Towards Deprescribing (rPATD) [26]; and

- Multimorbidity Treatment Burden Questionnaire (MTBQ) [27].

Healthcare utilisation data were collected to assess the effect of the intervention on healthcare usage and for the trial's economic evaluation.

Outcomes were collected at baseline and at 6 months after intervention delivery. Patient-reported measures were collected by postal questionnaires. Data for all other measures including prescribed medicines, medical and investigations history, and healthcare utilisation were collected by participating GPs and submitted to the study manager (CMC). This was a deviation from the original protocol, which indicated that these data would be collected by the research team. This deviation related to changes in data protection and national health research regulations during the study period, which precluded the research team access to the patients' full clinical record.

## Adverse events

Information on adverse events such as mortality, emergency department (ED) presentations, and hospital admissions was collected at follow-up. Given the deprescribing approach of the intervention, a safety protocol for identifying and reporting any suspected adverse drug withdrawal events (ADWEs) was developed. An ADWE is defined as either recurrence of the condition for which the drug was prescribed (e.g., recurrence of angina after stopping a beta blocker) or a physiologic reaction to drug withdrawal (e.g., selective serotonin reuptake inhibitor [SSRI] withdrawal syndrome) [28,29]. Although discontinuing medicines in older people has been demonstrated to be safe [30], given the paramount importance of the principle of "do no harm" in research ethics, a vigorous and detailed method was established to ensure that any potential ADWEs precipitated by deprescribing in a SPPiRE medication review were captured. Intervention GPs were asked to report any possible ADWE following the SPPiRE medication review. The Naranjo ADR probability scale [31] has been adapted in other studies to assess the likelihood that a reaction is related to drug withdrawal [28,29]. This tool was further adapted for SPPiRE and used to make an assessment on the causality of the ADWE. To ensure that the patient perspective was included, self-reported possible ADWEs were also collected from patient follow-up questionnaires.

## Patient and public involvement

There was no formal patient and public involvement (PPI) during the intervention development process. However, there was PPI representation on the independent trial steering committee that oversaw the completion of the trial, reviewed all trial documentation, and was involved in the development of the safety protocol and methods for monitoring adverse events.

## Sample size

As outlined in the trial protocol [22], the study was designed with 90% power to detect a 20% reduction in the proportion with PIP and a mean difference of one medicine between intervention and control groups (based on a mean of 17.4 medicines standard deviation [SD] 2.6) and the sample size inflated to incorporate the effects of clustering (using an intraclass correlation [ICC] of 0.025). The sample size was recalculated when it became apparent during early recruitment that it would not be possible to recruit clusters with an average of 15 participants, as was initially planned in the protocol. An average cluster size of 8 was anticipated, which inflated the original sample size from 30 practices (450 patients) to 50 practices (400 patients).

## Statistical analysis

Descriptive statistics were used to describe baseline characteristics of recruited practices and participants. All analyses were conducted under the intention-to-treat principle, and those lost to follow-up had their baseline data carried forward. The primary analysis was carried out using multilevel modelling. The first primary outcome measure, number of repeat medications, was assessed using mixed effects Poisson regression with the individual as the unit of analysis and the practice included as the random effect to control for the effects of clustering and results presented using incidence rate ratios (IRRs) and 95% confidence intervals (CIs). The baseline number of medicines, GP size (number of GP sessions per week), and GP location (urban/rural) were included in the analysis as fixed effects. The second outcome measure, proportion of patients with a PIP, was analysed in a similar manner using mixed effects logistic regression, including PIP at baseline, GP size and location, and results presented using odds

ratios (ORs) and 95% CIs. A number of prespecified sensitivity analyses were conducted: complete case analysis, per-protocol analysis, and including "presence of a repeat prescribing policy" as a covariate. All secondary outcomes were analysed in a similar manner to the primary outcomes, using appropriate mixed effects regression methods (i.e., linear, logistic, and Poisson).

## Results

### Practice and patient recruitment and retention

Between April 2017 and December 2019, 139 practices and 1,626 patients were invited to take part. A total of 51 practices were recruited giving an overall practice enrollment rate of 36.7% (see Fig 2).

Practices were excluded either because they were unable to identify (n = 23) or recruit (n = 23) a sufficient number of eligible patients. In some practices, the finder tool did not function as anticipated due to problems with how prescription data were coded in that practice. Of the practices who declined to take part, most cited time constraints as the primary reason. Practices recruited into the trial were larger compared to the national average, for example, 39% of SPPiRE practices had 4 GPs, compared to 19% nationally.

Of the 1,626 patients invited, 404 were ultimately recruited into the trial giving an enrollment rate of 24.8% (see Fig 2). A total of 18 patients (4.45%) were recruited and allocated but were subsequently withdrawn from the trial (9 from each arm), during an audit of consent forms, following changes in health research regulations, a process that was overseen by the trial's independent steering committee. Moreover, 35 patients (8.66%) were lost to follow-up, 21 of whom died during the study period. Participants lost to follow-up were older, had a higher number of medicines at baseline, a higher number of PIP, and lower mean EQVAS scores (see S3 Table).

### Baseline characteristics

Recruited patients had a mean age of 76.5 years (SD 6.83), a mean number of medicines of 17.37 (SD 3.50), and a mean number of PIP per person of 2.52 (SD 1.48). Between one quarter and one-third of participants scored in the severe or extreme domains of the EQ-5D-5L for impairments to mobility, activities of daily living, and pain at baseline. Practices and patients in each group had similar characteristics at baseline; see Table 1 for practice details and Table 2 for patient demographic details in each group.

With respect to primary outcome measures, the intervention group had lower number of medicines at baseline (16.96 compared to 17.82), although this was adjusted for in the analysis. The prevalence of PIP was similar in both groups. See S3 Table for primary outcomes in each group at baseline and S4 Table for secondary outcome measures in each group at baseline.

### Primary and secondary outcomes

All 404 patients were included in the intention-to-treat analysis of primary outcome measures. There was a reduction in the number of medicines and PIP in each group at follow-up (Table 3). With respect to the adjusted difference in the number of medicines in the intervention compared to control group at follow-up, there was a small but significant effect (IRR 0.95, 95% CI: 0.899 to 0.999, $p$ = 0.045). There was no evidence of an effect on the adjusted odds of having a PIP in the intervention group, compared to the control group, at follow-up (OR 0.39, 95% CI: 0.140 to 1.064, $p$ = 0.066). See S5 for sensitivity analyses, the results of which were similar to the intention-to-treat analysis.

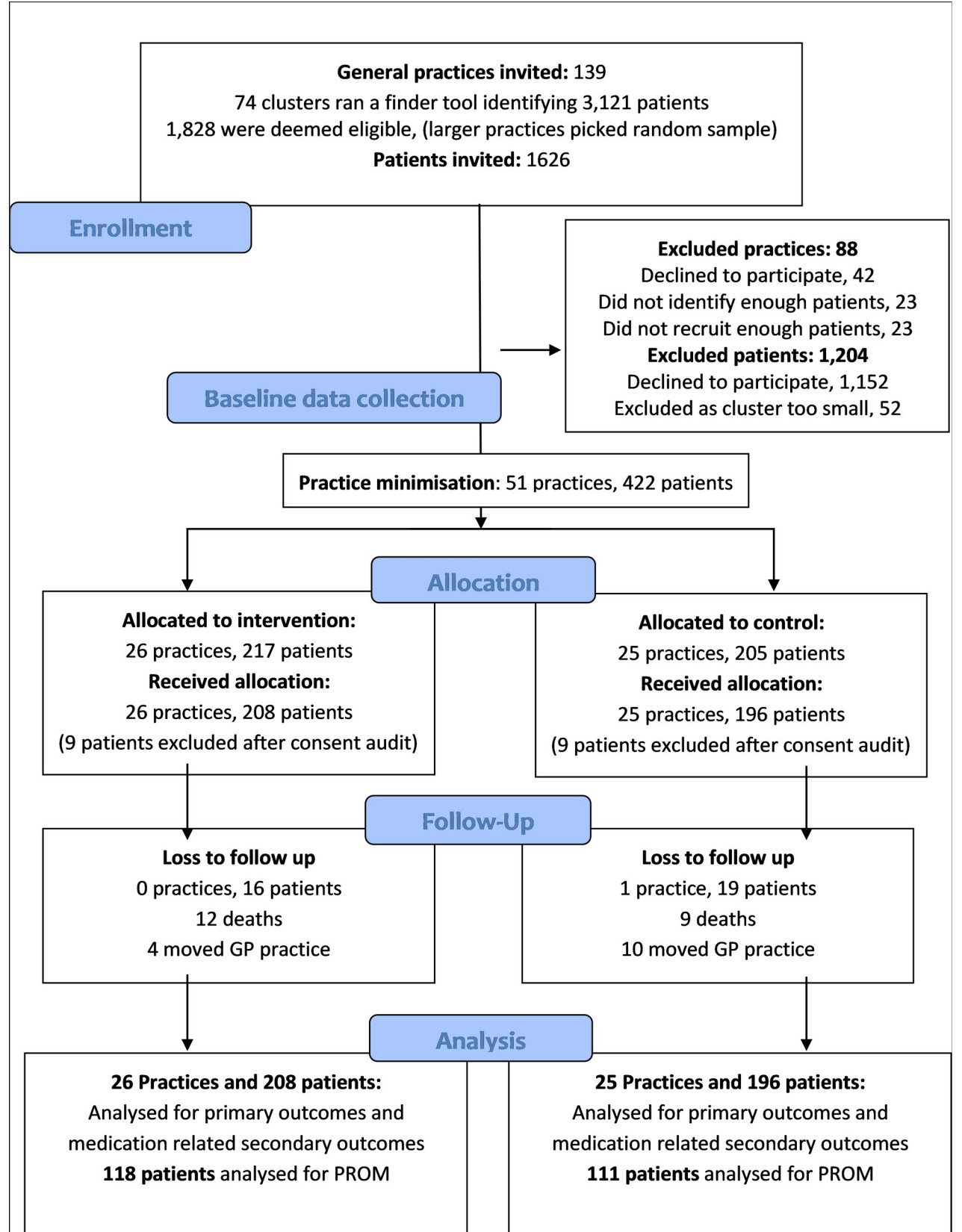

**Fig 2. SPPiRE trial participant flow diagram.** GP, general practitioner; PROM, patient-reported outcome measure; SPPiRE, Supporting Prescribing in Older Adults with Multimorbidity in Irish Primary Care.

With respect to secondary prescribing-related measures, there were significantly more medicines stopped and a significant reduction in the odds of being prescribed ≥15 medicines in the intervention compared to the control group at follow-up. There was no evidence of an effect demonstrated on any of the PIP-related or patient-reported outcome measures or on healthcare utilisation. There was a reduction in the number of GP visits and an increase in the number of telephone consultations in both groups at follow-up. Over a quarter of SPPiRE participants had follow-up dates after the outbreak of the Coronavirus Disease 2019 (COVID-19) pandemic in March 2020, and, unsurprisingly, this group had significantly less face-to-face GP visits and significantly more telephone consultations.

## Adverse events

There were 21 deaths during the study period: 9 in the control and 12 in the intervention group. None of the intervention deaths were reported as being related to the intervention. A total of 15 ADWEs out of 826 stopped drugs in intervention patients (1.81%) were reported by intervention group GPs and 10 of these events were assessed as being probably related to drug withdrawal (control group GPs did not collect these data as they continued to provide care as usual). One of the ADWEs was categorised as serious by the intervention GP, a severe depressive episode requiring inpatient admission 8 weeks after discontinuing a serotonin–norepinephrine reuptake inhibitor (SNRI). The remaining reactions were categorised as mild and all

**Table 1.  Practice details in each group at baseline.**

| Characteristic | Intervention, *n* = 26 | Control, *n* = 25 |
|---|---|---|
| *Number of GP sessions per week* | | |
| Mean (SD) | 30.42 (17.35) | 27.54 (13.78) |
| Median (IQR) | 29.50 (18 to 37) | 26 (16 to 35.5) |
| *Number of nurse sessions per week* | | |
| Mean (SD) | 12.40 (7.09) | 10.79 (5.68) |
| Median (IQR) | 10 (9 to 15) | 10 (7.5 to 12.5) |
| *Practice manager* | | |
| None (%) | 3 (11.5) | 6 (24.0) |
| Part time (%) | 9 (34.6) | 9 (36.0) |
| Full time (%) | 14 (53.8) | 10 (40.0) |
| *Total number of patients* | | |
| Mean (SD) | 6,877.72 (3,354.24) | 6,512.56 (3,942.18) |
| Median (IQR) | 6,850 (5,484 to 7,994) | 5,948 (3,265 to 8,519) |
| *Number of patients aged ≥65 years* | | |
| Mean (SD) | 1,192.78 (916.78) | 1,192.78 (650.66) |
| Median (IQR) | 974.5 (625 to 1,248) | 714 (591 to 1,422) |
| *Location* | | |
| Urban (%) | 14 (53.8) | 16 (64) |
| Rural (%) | 4 (15.4) | 2 (8) |
| Mixed (%) | 8 (30.8) | 7 (28) |
| Written repeat prescribing policy (%) | 14 (53.8) | 11 (44) |

IQR, interquartile range; SD, standard deviation.

**Table 2. Patient demographics in each group at baseline.**

| Patient demographic detail | Intervention (N = 208) | | Control (N = 196) | |
|---|---|---|---|---|
| **Age** | | | | |
| Mean (SD) | 76.67 (6.80) | | 76.33 (3.89) | |
| Median (IQR) | 76 (71 to 82) | | 76 (70 to 82) | |
| **Sex** | | | | |
| | *N* | % | *N* | % |
| Male | 89 | 42.79 | 84 | 42.86 |
| Female | 119 | 57.21 | 112 | 57.14 |
| **GMS card holder** | | | | |
| GMS card | 164 | 78.85 | 167 | 85.20 |
| No GMS card | 36 | 17.31 | 19 | 9.69 |
| Unknown | 8 | 3.85 | 10 | 5.10 |
| **Language** | | | | |
| Language other than English | 3 | 1.44 | 2 | 1.02 |
| English | 195 | 93.75 | 180 | 91.84 |
| Unknown | 11 | 5.29 | 14 | 7.14 |
| **Occupation** | | | | |
| Professional worker | 13 | 6.25 | 11 | 5.61 |
| Managerial and technical | 38 | 18.27 | 25 | 12.76 |
| Nonmanual | 30 | 14.42 | 26 | 13.27 |
| Skilled manual | 26 | 12.50 | 26 | 13.27 |
| Semiskilled | 11 | 5.29 | 18 | 9.18 |
| Unskilled | 9 | 4.33 | 7 | 3.57 |
| Farmer, size of farm unspecified | 5 | 2.40 | 10 | 5.10 |
| Unknown | 52 | 25.00 | 51 | 26.02 |
| Homemaker | 24 | 11.54 | 22 | 11.22 |
| **Education** | | | | |
| No schooling | 0 | 0.00 | 3 | 1.53 |
| Primary school education only | 69 | 33.17 | 86 | 43.88 |
| Some secondary education | 53 | 25.48 | 44 | 22.45 |
| Complete secondary education | 40 | 19.23 | 20 | 10.20 |
| Some third level education | 20 | 9.62 | 19 | 9.69 |
| Complete third level education | 16 | 7.69 | 13 | 6.63 |
| Unknown | 10 | 4.81 | 11 | 5.61 |
| **Employment** | | | | |
| Employed | 1 | 0.48 | 0 | 0.00 |
| Self employed | 7 | 3.37 | 4 | 2.04 |
| Retired | 156 | 75.00 | 146 | 74.49 |
| Homemaker | 32 | 15.38 | 31 | 15.82 |
| Other/unknown | 12 | 5.77 | 15 | 7.65 |

GMS, general medical services; IQR, interquartile range; SD, standard deviation.

resolved with reinstitution of the drug (examples included an itch after discontinuing an antihistamine and dyspepsia after discontinuing a proton pump inhibitor).

## Protocol adherence

Between January 2, 2018 and May 11, 2020, 163 of 208 (78.37%) intervention patients had a SPPiRE medication review. Due to individual practice circumstances, there were delays in

**Table 3. Primary and secondary outcome measures at follow-up.**

| Outcome measure | Intervention (N = 208) | Control (N = 196) | Adjusted difference (95% CI); p-value |
|---|---|---|---|
| **Primary outcome measures** | | | |
| Number of medicines¥, Mean (SD) | 16.02 (3.93) | 17.55 (4.10) | 0.95‡ (0.899 to 0.999); p = 0.045 |
| Patients with at least 1 PIP§, N (%) | 181 (87.44) | 179 (91.79) | 0.39¶ (0.140 to 1.064); p = 0.066 |
| **Secondary outcome measures** | | | |
| **Prescribing-related measures** | N = 208 | N = 196 | |
| Number of medicines stopped, Mean (SD) | 3.97 (3.15) | 2.92 (3.17) | 1.48‡ (1.171 to 1.871); p = 0.001 |
| Number of medicines started, Mean (SD) | 3.02 (3.03) | 2.67 (2.91) | 1.12‡ (0.826 to 1.513); p = 0.470 |
| Proportion prescribed ≥15 medicines, N (%) | 132 (63.46) | 161 (82.14) | 0.37¶ (0.193 to 0.719); p = 0.003 |
| Number of PIP, Mean (SD) | 2.16 (1.44) | 2.35 (1.43) | 0.92‡ (0.813 to 1.057); p = 0.256 |
| Proportion with any reduction in PIP, N (%) | 73 (35.10) | 58 (29.51) | 1.42¶ (0.892 to 2.255); p = 0.140 |
| Proportion with at least 1 high-risk PIP, N (%) | 117 (57.07) | 119 (62.30) | 0.93¶ (0.528 to 1.642); p = 0.806 |
| **PROMs** | N = 118 | N = 111 | |
| EQ-5D-5L index score, Mean (SD) | 0.517 (0.382) | 0.456 (0.357) | 0.011₱ (−0.059 to 0.081); p = 0.753 |
| Global MTBQ score, Mean (SD) [27] | 12.33 (13.50) | 17.05 (16.42) | −2.80₱ (−6.433 to 0.833); p = 0.131 |
| Satisfied with medicines (rPATD [26] question 1)*, Mean score (SD) | 4.21 (0.76) | 3.94 (0.76) | 1.06‡ (0.922 to 1.212); p = 0.427 |
| Willing to stop a medicine (rPATD [26] question 7)*, Mean score (SD) | 4.38 (0.71) | 4.11 (0.82) | 1.05‡ (0.918 to 1.200); p = 0.478 |
| **Healthcare utilisation** | N = 188 | N = 171 | |
| Number of GP visits, Mean (SD) | 4.42 (3.51) | 3.83 (3.26) | 1.06‡ (0.840 to 1.348); p = 0.608 |
| Telephone consultations, Mean (SD) | 1.55 (2.12) | 1.41 (2.21) | 1.75‡ (0.995 to 3.081); p = 0.052 |
| Repeat prescription requests, Mean (SD) | 2.51 (2.55) | 2.38 (2.57) | 1.08‡ (0.817 to 1.424); p = 0.593 |
| ED presentations, Mean (SD) | 0.46 (1.01) | 0.33 (0.85) | 1.31‡ (0.699 to 2.477); p = 0.394 |
| Number of inpatient days, Mean (SD) | 2.43 (6.19) | 3.07 (9.79) | 1.50‡ (0.522 to 4.308); p = 0.451 |
| Number of outpatient visits, Mean (SD) | 2.62 (5.54) | 2.39 (2.28) | 0.97‡ (0.712 to 1.333); p = 0.869 |

§ Intracluster correlation coefficient at baseline was 0.053 (95% CI: 0.000 to 0.122).

¥ Intracluster correlation coefficient at baseline was 0.171 (95% CI: 0.074 to 0.269).

‡ IRR from multilevel Poisson regression.

¶ OR from multilevel logistic regression.

₱ Beta coefficient from multilevel linear regression.

* Five-point Likert score where 1 represents strongly disagree and 5 strongly agree.

CI, confidence interval; ED, emergency department; EQ-5D-5L, EuroQoL 5-dimension 5-level; GP, general practitioner; IRR, incidence rate ratio; MTBQ, Multimorbidity Treatment Burden Questionnaire; OR, odds ratio; PIP, potentially inappropriate prescribing; PROM, patient-reported outcome measure; rPATD, revised Patients' Attitudes Towards Deprescribing; SD, standard deviation.

some intervention practices in completing SPPiRE reviews. The trial management committee took a flexible approach and allowed some additional time when requested by intervention practices. This led to a chance imbalance in the median number of days to follow-up between groups: 431 days (interquartile range [IQR] 366 to 494) in the control group and 557 days (IQR 418 to 627.5) in the intervention group. Due to this identified discrepancy, a sensitivity analysis was performed including number of days to follow-up as a covariate (see S5 Table); however, there was little difference in the results.

Of the 45 intervention patients (21.63%) that did not have a SPPiRE medication review, 36 patients did not have a review because of GP factors, primarily insufficient time. A total of 9 patients did not have a review as they had either died, moved to another GP practice, or were too unwell at the time of the review to attend the practice. There were 3 control practices who had a large reduction in the mean number of medicines per participant during the study period of 8.6, 5.4, and 3.5 medicines per person, respectively. See S5 Table for sensitivity analyses including a per-protocol analysis, which showed a slight reduction in treatment effect.

## Discussion

The SPPiRE intervention resulted in a significant reduction in the number of medicines in the intervention compared to the control group at follow-up. The impact of this reduction (equating to 0.85 medicines per person on average) on an individual patient is unclear. However, the intention with widespread implementation of this type of intervention would be ongoing medication reviews every 6 to 12 months, as opposed to a once off review, and that may lead to incremental improvements and deprescribing of unnecessary or inappropriate medicines over time. There is also a possibility that normalisation of deprescribing as a concept would in itself lead to a change in attitudes and behaviours among prescribers. Although the effect size was small, if implemented at scale, it would have a significant impact at a population level. One of the explanations for the small effect size was that there was also a reduction in the number of medicines in the control group during the study period. This is unexpected in the context of the trends observed in cross-sectional and prospective cohort studies that suggest polypharmacy increases in individuals as they age [32–36]. In addition, although the intervention did not demonstrate any significant effect on PIP, there were improvements in PIP in both groups. Although control practices were not provided with any of the intervention support material, it is possible that some performed medication reviews on recruited patients as they were aware of those recruited due to their role in screening patients for eligibility, indicating a potential Hawthorne effect in the control group. The mechanism of action of the intervention and the potential feasibility and cost-effectiveness of system wide implementation will be further assessed by the ongoing process and economic evaluations [22,37].

### Strengths and limitations

This pragmatic trial recruited to target a vulnerable group of patients with substantial disease and treatment burden and a high baseline prevalence of PIP, and the intervention was delivered in routine primary care with no other resource or material necessary aside from the SPPiRE website. In line with findings from international deprescribing trials, the SPPiRE intervention was both safe and feasible [38]. A rigorous protocol was developed to identify any potential adverse events relating to medication withdrawal. During the study period, there were 826 medicines stopped in the intervention group, with just 15 adverse events reported.

This study has some limitations. First, only a quarter of invited patients agreed to participate, and this is lower than other similar studies [39], although no other intervention study with this degree of polypharmacy as an inclusion criteria could be identified for comparison. In addition, only 45% of patients reported patient-reported outcome measures at follow-up (see S3 Table for a comparison of those with and without follow-up patient-reported outcome measure data). Second, outcome measures were assessed at just one time point 6 months after intervention completion as we wanted to assess the sustained effect of the intervention rather than examine an immediate intervention effect. Repeated measurement of prescribing outcomes would be ideal but was not possible in our health system due to the inability to electronically extract these data from practices. During the study period, 1,398 medicines were stopped and 1,153 medicines started, and over 80% of participants had at least 1 medicine stopped and started, indicating that prescribing for patients with this degree of polypharmacy is a complex process in constant flux. There is a possibility that the full effect of the intervention may not have been captured by assessing medication count and appropriateness at just one point in time. While the SPPiRE trial was ongoing, a secondary data analysis of the Prioritising Multi-medication in Multimorbidity (PRIMUM) trial by Muth and colleagues evaluating a similar medicines management multimorbidity intervention was published that raised important

questions about the appropriateness of cross-sectional analyses of prescriptions in these patients [40]. The rationale for the selection of the SPPiRE primary outcome measures is described in detail elsewhere [19], but one limitation of using medication count and explicit measures of medication appropriateness as outcome measures is that there was no assessment of the overall appropriateness of prescriptions.

Although returning broadly similar results, the per-protocol analysis showed an unexpected slight reduction in treatment effect. A total of 3 intervention practices that did not perform any reviews using the SPPiRE website showed a higher reduction in the number of medicines and PIP compared to the intervention group average. As these GPs had access to all intervention material, it is possible that they incorporated aspects of the intervention into their practice during the intervention period.

Finally, a chance imbalance in the number of days from baseline to follow-up between groups was identified. With respect to the impact on the primary outcomes, there is evidence that polypharmacy in this age cohort increases over time [41,42], which may have favoured the control group, which had a median of 126 less days to follow-up compared to the intervention group. However, a sensitivity analysis including the number of days to follow-up as a covariate revealed that there was no significant effect on the results.

## Comparison with similar medicines management interventions

Compared to other recently published studies of medicines management multimorbidity interventions, the SPPiRE intervention was not as intensive, involving a once off 30- to 40-minute review per patient with online educational material available for intervention GPs. In addition, patients recruited to the SPPiRE trial were on average older and with a notably higher mean number of medicines. While SPPiRE did have some effect, it may be that a more intensive approach is needed for patients with this degree of multimorbidity. However, 3 recently published multimorbidity intervention studies incorporated a GP-delivered medication review as part of a wider multidisciplinary delivered intervention [39,43,44], and, despite their more intensive interventions, none of these studies demonstrated a significant effect on their primary outcomes of health-related quality of life [40], the medication appropriateness index [43], and process indicators of intervention implementation [44]. The 3D study was the only of these 3 previous studies to report details on what proportion of intervention participants completed the intervention as planned, where 75% of participants (similar to SPPiRE) had at least one 3D review over the 15-month intervention period [39]. In addition, other healthcare professionals, such as pharmacists, were not directly involved in the SPPIRE intervention. The Pharmacist-Led Information Technology Intervention for Medication Errors (PINCER) trial demonstrated the clinical and cost-effectiveness of pharmacist-delivered medication reviews in primary care in the UK, although the intervention was more intensive and was conducted over time [45]. A small uncontrolled study based in Irish primary care has recently demonstrated the feasibility of pharmacist-delivered medication reviews [46]; however, given that SPPiRE was designed as a pragmatic nationwide RCT and that primary care pharmacists are not a part of routine care in Ireland, the SPPiRE intervention was designed as a GP-delivered medication review. Of note, none of the previously published trials specifically targeted this complex multimorbidity group.

With respect to polypharmacy studies, similar to SPPiRE, a systemic review looking at interventions to improve the appropriate use of polypharmacy in older people concluded that there was no consistent effect on PIP across the 32 included studies [47]. A meta-analysis of 25 deprescribing RCTs where the mean number of medicines at baseline was 7.4 showed a small but significant reduction in the mean number of medicines [48].

### Patient-reported outcome measures

Patient-reported outcome measures are used in multimorbidity studies to capture the patient's perspective of intervention effectiveness [49]. Although not powered to detect a change in these measures and hampered by the poor response rate to postal questionnaires at follow-up (57%), the SPPiRE intervention had no effect on any patient-reported outcome measure. Similar medicines management multimorbidity studies have also largely failed to demonstrate any effect on these measures. A total of 3 other studies have examined the effect of a medicines management intervention on health-related quality of life; 2 of these studies [39,50] did not show any effect on EQ-5D scores, while 1 demonstrated a significant increase in the EQVAS score, although no effect on the index score [51]. A total of 2 other medicines management multimorbidity studies used other health-related quality of life scales such as the SF36 and other patient-reported outcome measures such as the instrumental activities of daily living questionnaires [52,53]; neither study detected any difference between groups at follow-up. Use of patient-reported outcome measures in future multimorbidity intervention studies should be carefully considered given the likelihood, based on the low response rate, that potential SPPiRE participants were deterred from taking part due to difficulties in completing postal questionnaires.

## Conclusions

Due to our ageing population, the provision of safe, effective, and equitable healthcare for those with complex multimorbidity will become an ever pressing challenge. The SPPiRE trial demonstrated that a once off GP-delivered medication review had a small effect in reducing the number of medicines in older adults with significant polypharmacy, but no clear effect on the quality of prescribing. Patients with this degree of multimorbidity may be less amenable to the benefits of a once off medication review type intervention. The aims of the intervention to address PIP, unnecessary polypharmacy, and patient treatment priorities may have been over-ambitious given the degree of treatment burden in this population, which has not been extensively studied previously. Future medicines management multimorbidity studies should consider identifying patients who have some but not severe treatment burden using longitudinal assessments of medication-related outcome measures and carefully prioritising the use of patient-reported outcome measures. Although the effect size of the SPPiRE intervention was small and there was no clear effect demonstrated on the quality of prescribing, there is potential cost saving implications at a population level with system-wide implementation and deprescription of unnecessary medicines.

## Supporting information

**S1 Text. SPPiRE Study protocol.** SPPiRE, Supporting Prescribing in Older Adults with Multimorbidity in Irish Primary Care.
(DOCX)

**S1 Table. CONSORT checklist.** CONSORT, Consolidated Standards of Reporting Trials.
(DOCX)

**S2 Table. SPPiRE criteria.** SPPiRE, Supporting Prescribing in Older Adults with Multimorbidity in Irish Primary Care.
(DOCX)

**S3 Table. Comparison of participants followed up and lost to follow-up.**
(DOCX)

**S4 Table. Secondary outcome measures at baseline.**
(DOCX)

**S5 Table. Sensitivity analyses.**
(DOCX)

## Acknowledgments

We thank all the patients and GP practice staff who took part in this research. Membership of the wider SPPiRE Study group who contributed to this research (administration, pilot study, practice recruitment, data collection, data entry, and website development) were Tom Fahey, Derek Corrigan, Bridget Kiely, Aisling Croke, James Larkin, Oscar James, Clare Lambert, and Brenda Quigley. The independent members of the trial steering committee were Patricia Kearney (Chair), Andrew Murphy, Cathal Cadogan, Carmel Hughes, Paddy Gillespie, and Brid Nolan (PPI).

## Author Contributions

**Conceptualization:** Caroline McCarthy, Barbara Clyne, Fiona Boland, Frank Moriarty, Michelle Flood, Emma Wallace, Susan M. Smith.

**Data curation:** Caroline McCarthy, Barbara Clyne, Fiona Boland, Frank Moriarty, Emma Wallace, Susan M. Smith.

**Formal analysis:** Caroline McCarthy, Barbara Clyne, Fiona Boland, Frank Moriarty, Emma Wallace, Susan M. Smith.

**Funding acquisition:** Barbara Clyne, Emma Wallace, Susan M. Smith.

**Investigation:** Caroline McCarthy, Barbara Clyne, Frank Moriarty, Michelle Flood, Emma Wallace, Susan M. Smith.

**Methodology:** Caroline McCarthy, Barbara Clyne, Fiona Boland, Frank Moriarty, Emma Wallace, Susan M. Smith.

**Project administration:** Caroline McCarthy, Barbara Clyne, Susan M. Smith.

**Resources:** Caroline McCarthy, Barbara Clyne, Susan M. Smith.

**Supervision:** Barbara Clyne, Fiona Boland, Frank Moriarty, Michelle Flood, Emma Wallace, Susan M. Smith.

**Validation:** Caroline McCarthy, Barbara Clyne, Fiona Boland, Frank Moriarty, Emma Wallace, Susan M. Smith.

**Visualization:** Caroline McCarthy, Barbara Clyne, Fiona Boland, Frank Moriarty, Emma Wallace, Susan M. Smith.

**Writing – original draft:** Caroline McCarthy.

**Writing – review & editing:** Caroline McCarthy, Barbara Clyne, Fiona Boland, Frank Moriarty, Michelle Flood, Emma Wallace, Susan M. Smith.

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
