## [Editor Report · Decision Letter 0]

24 May 2021

Dear Dr McCarthy, 

Thank you for submitting your manuscript entitled "Effectiveness of a GP delivered medication review in reducing polypharmacy and potentially inappropriate prescribing in older patients with multimorbidity in Irish primary care: a cluster randomised controlled trial (SPPiRE study)" for consideration by PLOS Medicine.

Your manuscript has now been evaluated by the PLOS Medicine editorial staff and I am writing to let you know that we would like to send your submission out for external peer review.

Please re-submit your manuscript within two working days, i.e. by May 26 2021 11:59PM.

Kind regards,

Beryne Odeny

Associate Editor

PLOS Medicine

---

## [Decision Letter · Decision Letter 1]

2 Aug 2021

Dear Dr. McCarthy,

Thank you very much for submitting your manuscript "Effectiveness of a GP delivered medication review in reducing polypharmacy and potentially inappropriate prescribing in older patients with multimorbidity in Irish primary care: a cluster randomised controlled trial (SPPiRE study)" (PMEDICINE-D-21-02234R1) for consideration at PLOS Medicine. 

[LINK]

In light of these reviews, I am afraid that we will not be able to accept the manuscript for publication in the journal in its current form, but we would like to consider a revised version that addresses the reviewers' and editors' comments. Obviously we cannot make any decision about publication until we have seen the revised manuscript and your response, and we plan to seek re-review by one or more of the reviewers. 

We expect to receive your revised manuscript by Aug 23 2021 11:59PM. Please email us (plosmedicine@plos.org) if you have any questions or concerns.

We look forward to receiving your revised manuscript. 

Sincerely,

Beryne Odeny, 

PLOS Medicine 

plosmedicine.org

1. Abstract:

a) Please ensure that all numbers presented in the abstract are present and identical to numbers presented in the main manuscript text.

b) Please provide the number of GP practices in each group (i.e., control and intervention groups)

c) Please indicates the dates during which study enrollment and follow up occurred.

d) Please include a summary of adverse events in the study.

e) In the last sentence of the Abstract Methods and Findings section, please describe the main limitation(s) of the study's methodology.

2. Please include the study protocol document and analysis plan, with any amendments, as Supporting Information to be published with the manuscript if accepted.

3. Please merge the Aims section (line #111-115) with the Introduction section. Please conclude the Introduction with a clear description of the study question or hypothesis. 

4. In the Methods section’s text, please indicate the number of GP practices in each arm.

5. When completing the CONSORT checklist, please use section and paragraph numbers.

6. The terms gender and sex are not interchangeable (as discussed in http://www.who.int/gender/whatisgender/en/ ); please use the appropriate term.

7. Please replace "Social class" with "occupation" or similar throughout the paper.

8. Please use the PLOS Medicine style reference call outs throughout the text, noting the absence of spaces within the square brackets, e.g., "... quality of life [1-3]."

Comments from the reviewers:

Reviewer #1: The authors are to be congratulated on completing this cluster RCT in primary care - such trials are hard to do logistically and take a lot of effort. As others have found, including myself, the number of recruitable GP practices (only about a third) and the numbers of recruited patients (about a quarter of those screened) are always less than one hopes at the outset and clearly introduces the potential for selection bias and loss of generalisability, and attenuates the intervention effect as it is likely that GPs more interested in deprescribing specifically and research more generally may express interest in participating but whose practices are already at a more optimal level than those who do not. Similarly control patients who consented to participation, may be more open to medication review and deprescribing, coupled with the inevitable Hawthorne effect in control practices, as was seen in this study (and ours). The fact that about 20% of intervention effects did not receive the medication review, because of limited GP time or patient frailty/illness/death did not help either. The one-off nature of the intervention is a further limitation - could the authors make further comment about whether any subsequent GP consultations involved any further discussions around deprescribing. I note further process and qualitative studies are in progress but it would be interesting to see how easy GPs regarded the interaction with the SPPiRE website. While the STOPP criteria are validated they may be difficult to apply in individual cases because no set of rules will cover all possible scenarios, but also they are a limited set of rules and may not cover all potential deprescribing scenarios. I find it intriguing that intervention GPs who chose not to refer to the template but still undertook a deprescribing consultation achieved greater reduction in medications than those who did use the template and this may be explained by the former GPs internalising deprescribing concepts and applying these more generically to more drugs. This may in the end be a more effective approach than GPs referring to, and perhaps being constrained by, a pre-specified drug-class specific template and deserves further study. The loss of 45% of patients reporting PROMs is a further limitation and again introduces selection bias unless the authors can include as an appendix a comparison of characteristics of patients who did or did not report PROMs. But these limitations do not detract from the basic message that other similar studies have shown (including our own, which the authors may want to reference: Anderson K et al J Am Geriatr Soc 2020: 68:403-410), that deprescribing is not easy to do in time-pressured general practice, that the reductions in number of medication at the individual patient level are marginal but still worthwhile and do not cause harm, that follow-up consultations and longer follow-up may realise greater effects, and that remuneration policies for GPs specific to deprescribing or medication reviews are necessary if these tasks are to be performed adequately. 

Minor point - I would put table 2 as the first table because baseline characteristics of both groups ideally come first and place current table 1 as an appendix as I found it distracting and does not add much to what current table 2 shows. Also in appendix 4, it was unclear to me what the two sets of sensitivity analyses related to as they have identical headings and the same column variables although the data in the cells are slightly different and the IRRs are very different so the headings need further elaboration or at least a footnote. 

Reviewer #2: This cluster randomised controlled trial (SPPiRE) conducted in the Republic of Ireland aims to investigate the effect of a GP-delivered, individualised medication review in reducing polypharmacy and potentially inappropriate prescribing in older patients with multimorbidity in primary care.

Comments:

Can the authors please provide a copy of the study protocol in the supplementary material?

The authors have appropriately acknowledged where the original protocol was not adhered to, and transparently discussed the reasons for deviation within the manuscript. 

This includes a change in data collection methods and an updated sample size calculation.

Furthermore, the authors clearly and commendably provide a Protocol Adherence section, with appropriate sensitivity analyses and associated acknowledgement in the study limitations.

The CONSORT checklist has been suitably provided in the supplementary material.

The authors demonstrate rigor in the randomisation approach conducted and their attempts to minimise bias throughout the study design. 

Appropriate sensitivity analyses have been completed, including a complete case analysis and a per protocol analysis, which help to demonstrate the robustness of the study findings.

A valid modelling methodology has been applied for the study design, data structure, and research question in hand.

Overall, this is a well written and clearly presented study, with accurately and fairly communicated study outcomes, and adequately explored study limitations in the discussion section.

A small note: "All 404 patients were included in the intention to treat analysis of primary outcome measures, see 277 Error! Reference source not found.." to be rectified under the Primary and Secondary Outcomes section.

Reviewer #3: Thank you for the opportunity to review this manuscript. The SPPiRE study presents the results of a cluster randomised controlled trial of a GP delivered medication review where the intervention GPs received a web-based education module and template for medication review. The target population was patients aged 65 years or older and with what the authors define as "complex multimorbidity, defined as being prescribed ≥15 regular medicines".

The manuscript is well written, of interest to a wide readership, and the study itself is well conducted and reported.

I do have some comments and thoughts for the authors to consider and address:

1. The definition of polypharmacy in this study is not one that is widely accepted. Can you please reference why polypharmacy was defined as 15 or more medications? Polypharmacy is most commonly defined as 5 or more medications, and the other common term is hyperpolypharmacy referring to 10 or more medications. Need to please reference your polypharmacy definition of 15 or more meds. Has implications for whether this was an appropriate target outcome.

2. Polypharmacy and multimorbidity are not the same. Why was multimorbidity defined using the criteria of polypharmacy (15 or more meds)? 

Further, patients may have one medical disease e.g. heart failure and be on 5 medications for it's treatment so has implications for your study treatment effect. Please comment. 

3. Would also suggest avoiding arbitrary or emotive adjectives such as "extreme level of polypharmacy" (line 84) or "significant" polypharmacy (line 175) when talking about an outcome measure or inclusion criteria. Ok to use such words in narrative writing such as in the intro etc. Unless of course this is a formal measure that you can reference. But no citations have been provided as per comment 1 above.

4. Could you please provide a definition of repeat medication (line 227)?

5. The elephant in the room is COVID-19 - worth some discussion about how you think COVID-19 affected your study. 

6. Do you have data to talk about which PIPs you impacted on? The evidence for deprescribing is stronger for specific classes of medications.

7. I felt the Conclusion was introducing new concepts that were not presented in the Discussion. The discussion talks about reasons for the study intervention not being effective included its less intensive nature, but the Conclusion talks about the complexity of patients as being the potential reason for no effect. Can you tie these together?

8. Given your definition of multimorbidity was actually polypharmacy of 15 or meds. Can you compare your study to other trials that have targeted hyperpolypharmacy? 

9. Line 155 "discuss and agree changes" - is there a word missing before changes?

10. To reduce word count and repetitiveness the first few paragraphs of the Discussion as well the Strengths and limitations is all focussed on trying to explain your results - this could be truncated to then include comparisons with other relevant studies as per Comment 8.

Reviewer #4: Thank you for this article on a very important topic for older adults facing multimorbidity and polypharmacy. My comments are relatively minor but hope to address the international audience and also the multifaceted nature of the polypharmacy problem. 

Line 24: What does SPPiRE stand for? I know that this isn't the initial work coming from this intervention but if you could define the acronym perhaps in the abstract this would help frame the idea of the activity more clearly. 

Line 25: If this is an international audience then at some point also spelling out GP (which I am assuming is a general practitioner, or a primary care provider) would make sense. 

Line 80-86: I appreciate your description of necessary polypharmacy and your discussion of quality of prescriptions since I feel that polypharmacy is not a "black and white" issue and that several studies treat it as such and focus on numbers rather than individualized treatment plans. 

Line 135: given that these are community dwelling older adults based on your exclusion criteria you might mention that earlier in the manuscript, as homebound or nursing home patients might be more prone to polypharmacy and this study does not focus on them. 

Line 144: thank you for discussing the issues in blinding in this particular study and your attempts to blind the outcome measures. 

Line 149: Was there a way to assess whether intervention GPs actually played or watched the videos? Were logins tracked or some way to measure that the intervention was being deployed? 

Line 188: Given the deviation in data collection with the medical history do you think that the lack of standardization of the medical history might have impacted the discussion of necessary polypharmacy? Was there a process that the study manager tried to standardize these data that they received from the practices?

Line 346: Interesting discussion of the possible Hawthorne effect having some bearing on the study. 

Line 350-356: I think that this would be a very good study to undertake, as I had some concerns about the "independent study" nature of the intervention and whether the GPs were able to view and incorporate the material fully. This might also help refine the curriculum to the most high yield points (especially since it was noted that a limiting factor in participation was time constraints) 

In general I would like to see more discussion about the medical issues between groups and if the polypharmacy being addressed was deemed unnecessary or necessary (and if this was not able to be measured to have it be discussed in limitations, perhaps, since it is discussed in the introduction and quality of prescribing is also addressed in the conclusion)

[LINK]

---

## [Decision Letter · Decision Letter 2]

2 Nov 2021

Dear Dr. McCarthy,

Thank you very much for re-submitting your manuscript "Effectiveness of a GP delivered medication review in reducing polypharmacy and potentially inappropriate prescribing in older patients with multimorbidity in Irish primary care: a cluster randomised controlled trial (SPPiRE study)" (PMEDICINE-D-21-02234R2) for review by PLOS Medicine.

I have discussed the paper with my colleagues and the academic editor and it was also seen again by three reviewers. I am pleased to say that provided the remaining editorial and production issues are dealt with we are planning to accept the paper for publication in the journal.

[LINK]

We look forward to receiving the revised manuscript by Nov 09 2021 11:59PM.   

Sincerely,

Beryne Odeny,  

PLOS Medicine

plosmedicine.org

Requests from Editors: 

1) Please revise your title. Your title must be non-declarative and not a question. You may consider the following, or similar. Ideally only the study design should be in the sidetitle (i.e., after a colon). For example, “GP delivered medication review, polypharmacy and potentially inappropriate prescribing in older patients with multimorbidity in Irish primary care (SPPiRE study): a cluster randomised controlled trial”

Comments from Reviewers:

Reviewer #1: I am satisfied with the responses from the authors to my previous comments and welcome publication of the revised manuscript.

Reviewer #3: Thank you for considering and addressing all comments.

Reviewer #4: Thank you for this manuscript and for your attention to reviewer comments. I think that you have answered all of my questions regarding the study

[LINK]

---

## [Editor Report · Decision Letter 3]

5 Nov 2021

Dear Dr McCarthy, 

On behalf of my colleagues and the Academic Editor, Dr. Ian Scott, I am pleased to inform you that we have agreed to publish your manuscript "GP-delivered medication review of polypharmacy, deprescribing and patient priorities in older people with multimorbidity in Irish primary care (SPPiRE study): a cluster randomised controlled trial" (PMEDICINE-D-21-02234R3) in PLOS Medicine.

PUBLICATION SCHEDULE

Given our busy publication schedule for the remainder of 2021, we are planning to publish your paper in early January 2022 (the exact date will be communicated to you once confirmed).

PRESS

Sincerely, 

Beryne Odeny 

PLOS Medicine